# PD-L1 Silencing in Liver Using siRNAs Enhances Efficacy of Therapeutic Vaccination for Chronic Hepatitis B

**DOI:** 10.3390/biom12030470

**Published:** 2022-03-18

**Authors:** Till Bunse, Anna D. Kosinska, Thomas Michler, Ulrike Protzer

**Affiliations:** 1Institute of Virology, School of Medicine, Technical University of Munich/Helmholtz Zentrum München, Trogerstrasse 30, 81675 Munich, Germany; till.bunse@tum.de (T.B.); anna.kosinska@tum.de (A.D.K.); thomas.michler@tum.de (T.M.); 2German Center for Infection Research (DZIF), Partner Site Munich, 81675 Munich, Germany

**Keywords:** HBV, hepatitis, PD-L1, RNAi, therapeutic vaccination, immunology, checkpoint inhibition

## Abstract

In chronic hepatitis B virus (HBV) infection, virus-specific T cells are scarce and partially dysfunctional. Therapeutic vaccination is a promising strategy to induce and activate new virus-specific T cells. In long-term or high-level HBV carriers, however, therapeutic vaccination by itself may not suffice to cure HBV. One reason is the impairment of antiviral T cells by immune checkpoints. In this study, we used small-interfering RNA (siRNA) in combination with a heterologous prime-boost therapeutic vaccination scheme (*TherVacB*) to interfere with a major immune checkpoint, the interaction of programmed death protein-1 (PD-1) and its ligand (PDL-1). In mice persistently replicating HBV after infection with an adeno-associated virus harboring the HBV genome, siRNA targeting PD-L1 resulted in a higher functionality of HBV-specific CD8+ T cells after therapeutic vaccination, and allowed for a more sustained antiviral effect and control of HBV in peripheral blood and in the liver. The antiviral effect was more pronounced if PD-L1 was down-regulated during prime than during boost vaccination. Thus, targeting PD-L1 using siRNA is a promising approach to enhance the efficacy of therapeutic vaccination and finally cure HBV.

## 1. Introduction

Persistent infection with hepatitis B virus (HBV) and its accompanying disease, chronic hepatitis B, remain a global threat affecting 296 million people worldwide and causing an estimated 820,000 deaths per year [1]. Current standard of care using nucleoside analogues suppresses HBV replication, but seldomly eradicates the virus, and although the number of patients at risk to develop liver cirrhosis and hepatocellular carcinoma is reduced it remains high compared to healthy controls [2]. Furthermore, long-term drug administration poses challenges for clinical management, including drug distribution, high costs, adverse events or compliance issues. Thus, new treatment options which lead to cure or immune control of viral replication and by that allow treatment withdrawal are urgently needed. As the course of HBV infection is highly dependent on the antiviral immune response mounted by the patient, curative treatment strategies most likely need to activate antiviral immunity to control infection [3,4].

Several new immunotherapeutic approaches have emerged, one of the most potent being therapeutic vaccination. Clinical studies evaluating therapeutic vaccines for chronic hepatitis B patients, however, showed limited efficacy so far [5,6]. We previously developed a vaccination strategy named *TherVacB* that follows a heterologous prime-boost scheme [7]. Two protein-prime injections using a mixture of the two particulate recombinant protein antigens hepatitis B surface antigen (HBsAg) and HBV core antigen (HBcAg), mixed with a T cell activating adjuvant, are followed by a booster with a modified Vaccinia virus Ankara (MVA) expressing these antigens.

*TherVacB* showed promising results in preclinical mouse models of chronic hepatitis B [8], but recent findings indicate that immune checkpoints [9] as well as high antigenemia [8] inhibit vaccine-induced T cells. Thus, either lowering the antigen level or co-treatment with immune checkpoint inhibitors are promising approaches to enhance the efficacy of therapeutic vaccination [10]. Overexpression of co-inhibitory molecules, in particular PD-1, on circulating T cells in chronic hepatitis B patients highlights the importance of these molecules in persistent infection and associated T cell exhaustion [5,11,12,13]. The inhibitory PD-1/PD-L1 pathway is known to play a crucial role in the clearance of viral infections and is therefore an interesting target for therapies aimed to augment the immune response [14].

Previous studies showed that antibody-mediated PD-1/PD-L1 blockade could enhance the outcome of therapeutic vaccination in chronic infection with the woodchuck hepatitis virus, a virus closely related to HBV [12,15]. However, the use of an antibody to interfere with the PD-1/PD-L1 pathway comes with certain drawbacks. Due to the systemic effects of a blocking antibody, patients are at risk for adverse events such as diarrhea, colitis or hepatitis, caused by an unwanted activation of immune responses [16]. Thus, a targeted activation of T cells at the side of infection is preferable. An alternative to the systemic application of antibodies could be the use of RNA interference (RNAi). RNAi could down-regulate expression of inhibitory ligands in target organs such as the liver or even in certain cell types [17]. Combined with ligand-based or lipid nanoparticle-based delivery, organ or even cell-type specific immunomodulation can be achieved, reducing the risk of systemic adverse events [18].

Considering this, we investigated the effects of PD-L1 silencing, either mediated by RNAi or by a PD-L1-specific antibody on antiviral immunity and the course of HBV infection. We found that PD-L1 silencing enhanced antiviral immune responses especially when combined with therapeutic vaccination, which was even more pronounced by the liver-targeted siRNA compared to the antibody approach.

## 2. Materials and Methods

Animal Experimentation: Animal experiments were conducted in strict accordance with the German regulations of the Society for Laboratory Animal Science and the European Health Law of the Federation of Laboratory Animal Science Associations. Experiments were approved by the local Animal Care and Use Committee of Upper Bavaria (permission no. ROB-55.2-2532.Vet_02-18-24) and followed the 3R rules. Mice were kept in a specific pathogen-free facility under appropriate biosafety level following institutional guidelines. According to animal protection law, each experiment was performed once.

In vivo infection: Young (8–12 weeks old) C57BL/6J mice were infected by intravenous injection (i.v.) using an adeno-associated virus vector (AAV) serotype 2 genome encoding a 1.2-fold overlength of the HBV (genotype D) genome (and pseudotyped with an AAV serotype 8 capsid (AAV-HBV)). Persistent HBV replication was confirmed by HBV antigen measurements prior to the start of the experiments.

Therapeutic hepatitis B vaccination (*TherVacB*): Immunization was performed by biweekly intramuscular injections in the thigh. At week 0 and 2, 15 µg each of particulate HBsAg (genotype A, serotype adw) and HBcore antigen (genotype D, kindly provided by APP Latvijas Biomedicinas, Riga, Latvia) were mixed with 10 µg cyclic di-adenylate monophosphate(c-di-AMP) as an adjuvant directly before application; at week 4, injection of a recombinant MVA expressing HBV S and HBV core protein was performed (both genotype D) at a dose of 5 × 10^7^ collective infectious units (CIUs).

Analysis of HBV replication: Serum HsaBsAg, HBeAg, anti-HBs and anti-HBe levels were quantified using the ARCHITECT platform (Abbott, Libertyville Township, IL, USA). HBV RNA was isolated from whole liver lysates using a NucleoSpin RNA kit (Macherey-Nagel, Düren, Germany) and reverse transcribed using the SuperScript III Kit (Thermo Fisher Scientific, Waltham, MA, USA). HBV DNA was extracted from 50 µL serum using a High Pure Viral Nucleic Acid Kit (Roche Diagnostics, Rotkreuz, Switzerland). HBV pregenomic 3.5 kb RNA (pgRNA) was quantified by qPCR on a LightCycler 480 Instrument II (Roche Diagnostics, Rotkreuz, Switzerland) normalized for GAPDH or quantified using a cell-culture-derived HBV standard [19]. See Table 1 for primers and conditions.

Histology and immunohistochemistry: Livers were fixed using a 4% buffered formalin solution, dehydrated, and embedded in paraffin. After preparation of serial 2 µm thin sections, immunohistochemistry staining of HBcore was performed using rabbit anti-HBV core (#RP 017; 1:50 dilution; retrieval at 100 °C for 30 min with EDTA; Diagnostic Biosystems, Pleasanton, CA, USA). These slides were scanned and positive cells quantified from at least ten different sections.

Checkpoint inhibitor therapy: For antibody therapy, 100 µg rat anti-mPD-L1 (#BE0101; BioXCell, Lebanon, NH, USA) or the respective isotype control (#BE0090; BioXCell) were injected each day that a therapeutic vaccine component was given. For RNAi treatment, mice received 1 mg/kg body weight siRNA formulated with a transfection kit for targeting liver cells (Altogen Biosystems, Las Vegas, NV, USA) according to the manufacture’s protocol. PD-L1-specific (#4457308; Thermo Fisher Scientific, Waltham, MA, USA) or non-specific siRNA (#4459405) were administered i.v. 24 h prior to protein or MVA vaccination, respectively.

Intracellular cytokine staining (ICS) and detection of HBV-specific T cells:

Liver-associated lymphocytes (LAL) were isolated and purified by density gradient centrifugation as previously described [7], and surface staining was performed using anti-CD8 (clone 56.7-7; eBioscience, Frankfurt a. Main, Germany) and anti-CD4 (clone L3T4; BD Biosciences, Krefeld, Germany) antibodies. Dead cells were excluded by staining with fixable viability dye eF780 (eBioscience, Frankfurt a. Main, Germany). For intracellular cytokine staining, LAL were stimulated using peptide pools derived from HBs and HBcore for 16 h overnight in the presence of Brefeldin A. Subsequentially, cell permeabilization (Fixation/Permeabilization Solution Kit, BD Bioscience, Franklin Lakes, NJ, USA) and ICS was performed using antibodies against IFNγ (clone XMG1.2; BD Biosciences), TNFα (clone MP6-XT22; BD Biosciences, Franklin Lakes, NJ, USA) and granzyme B (clone GB11, Invitrogen, Waltham, MA, USA).

Major histocompatibility complex class I multimers were used for detection of HBV-specific T cells. Multimers were biotinylated and conjugated with Kb-restricted HBV-derived peptides S_190–197_ (S_190_; VWLSAIWM) and core_93–100_ (C_93_; MGLKFRQL), MVA-derived peptide B8R_20–27_ (MVA_B8R_; TSYKFESV) and as control ovalbumin-derived peptide S8L_257–264_ (OVA; SIINFEKL) (kindly provided by Dirk Busch, Technical University of Munich, Germany). Strep-Tactin labelled Streptamers were added to cell surface antibody mix, containing antibodies against PD-1 (clone J43; Invitrogen), LAG-3 (clone C9B7W; Invitrogen) and anti-TIM3 (clone B8.2C12; Biolegend, San Diego, CA, USA). 

## 3. Results

### 3.1. A Knock-Down of PD-L1 in the Liver by siRNA Improves T Cell Functionality

In a first step, we evaluated if siRNA-mediated silencing of PD-L1 in liver tissue would enhance T cell responses induced by therapeutic vaccination. An antibody-mediated blockade of PD-L1 served as control. For this, we used a previously described chronic hepatitis B mouse model, which is based on an infection with a recombinant adeno-associated virus (AAV) harboring a 1.2. overlength of the HBV genome (AAV-HBV), inducing persistent HBV replication when injected intravenously [20]. Persistent infection was confirmed by measuring HBV antigen levels (hepatitis B surface (HBsAg) and e antigen (HBeAg)) in the serum. Nine weeks after transduction, mice were immunized according to our *TherVacB* scheme in biweekly intervals: two injections of particulate HBsAg and HBcAg mixed with c-diAMP as adjuvant activating the stimulator of interferon genes (STING), followed by one injection of an MVA expressing HBV S and core proteins. In addition, mice received either siRNA targeting PD-L1 (siPD-L1), a rat monoclonal antibody against murine PD-L1 (anti-PD-L1), or the respective control siRNA (siCtrl) or isotype control antibody (AbCtrl) (Figure 1A). Two weeks after MVA boost, mice were sacrificed and the HBV-specific T cell response was analyzed.

Based on the assumption that a permanent antigen stimulation due to a chronic viral infection may lead to higher expression levels of co-inhibitory molecules on pathogen-specific T cells, we first characterized HBV-specific T cells in comparison to MVA-specific T cells (Figure 1B). The vast majority of T cells specific for the HBV C_93_ peptide expressed two or even three inhibitory molecules, i.e., PD-1, LAG-3 or TIM3, while most T cells specific for B8R, an MVA-derived peptide [21], expressed none or only one. 

As the co-expression of several inhibitory molecules is commonly accompanied by a decreased effector function and then referred to as T cell exhaustion, we next analyzed T cell functionality. Indeed, when we restimulated LAL with peptide pools covering the complete HBV S- or core protein and analyzed T cell activation by intracellular cytokine staining (ICS), we found only few reactive T cells after *TherVacB* vaccination when mice had received siCtrl (Figure 1C,D). In contrast, mice treated with siPD-L1 showed significantly higher expression of granzyme B, interferon (IFN)γ and tumor necrosis factor (TNFα), indicating increased T cell functionality.

To compare the effect of siRNA and antibodies targeting PD-L1, mice were treated with either siPD-L1, siCtrl, anti-PD-L1 or AbCtrl on the day before each vaccination in an independent experiment and LAL isolated two weeks after booster vaccination with MVA. ICS revealed that blocking PD-L1 by a checkpoint inhibitor antibody as well as down-regulating PD-L1 by siRNA increased the percentage of cytotoxic, granzyme B-releasing CD8^+^ T cells compared to the respective control. Here, siRNA proved superior over antibody treatment in activating S-specific T cells, with the same trend observed for core-specific T cells.

### 3.2. T Cells Activated by Therapeutic Vaccination Show an Improved Antiviral Effect after siRNA-Mediated Knock-Down of PD-L1

We next investigated whether the improved T cell functionality translates into a stronger antiviral effect. For this, we repeated the experiment, but this time also included treatment groups that received siRNA only but not vaccination, to investigate whether silencing of PD-L1 would already be sufficient to allow a spontaneous recovery of antiviral immunity.

Following vaccination of the mice with *TherVacB*, we detected a decrease in HBsAg and HBeAg serum levels (Figure 2A,B). This was accompanied by an induction of an antibody response against the respective antigen (Figure 2C,D), which was not observed in any of the non-vaccinated mice irrespective of whether they received siPD-L1 or not. Mice which were pre-treated with siPD-L1 prior to vaccination showed a more pronounced reduction of HBeAg and a complete loss of HBsAg. ALT activity was slightly elevated in mice receiving siPD-L1 although this treatment had no antiviral effect, indicating some level of hepatocyte toxicity (Figure 2E). ALT was significantly elevated in mice which had received siPD-L1 before therapeutic vaccination and showed a sustained antiviral effect, indicating that that the loss of viral antigens was at least partially due to a killing of HBV-replicating hepatocytes. This diminished replication was also reflected by reduced HBV DNA levels in the serum (Figure 2F) of vaccinated mice. 

To investigate how the different treatment combinations affected HBV replication in the liver, we quantified HBV pregenomic RNA (pgRNA) as a maker for production of new virus in liver lysates by qPCR and determined the number cells staining positive for HBV core by immunohistochemical staining. Those animals which had received *TherVacB* in combination with siPD-L1 showed a significant reduction in pgRNA (Figure 3A), indicating a reduced viral replication in the liver. Confirming this finding, the respective mice almost completely cleared HBV-positive hepatocytes from the liver (Figure 3B,C).

These findings confirm the results obtained from serum analysis, indicating that siRNA-mediated knock-down of PD-L1 in the liver can potentiate the effects of therapeutic hepatitis B vaccination. A knock-down of PD-L1 on its own, however, did not show any beneficial effect despite still bearing the risk of liver toxicity.

### 3.3. PD-L1 Knock-Down in the Liver Elicits Its Main Effect during Prime and Not during Boost Vaccination

Next, we wanted to decipher during which step of immune activation by therapeutic vaccination checkpoint inhibition is most critical. We therefore designed an experiment in which mice received siPD-L1 or siCtrl either during the priming or the boosting phase of the vaccination regimen. By priming with adjuvanted particulate protein vaccine components, *TherVacB* mainly activates B cells and CD4 T cells. In contrast, the booster using an MVA vector mainly expands and strengthens T cell responses [7,8]. 

Mice receiving *TherVacB* in combination with siCtrl showed a 1-log drop in HBsAg and an approx. 50% reduction in HBeAg during initial protein-prime vaccination, but no further drop after MVA boost vaccination (Figure 4A,B). HBsAg was cleared in all mice receiving siPD-L1 irrespective of when the siRNA was applied. In line with this, all groups of mice developed comparable anti-HB titers (Figure 4A,C). However, HBeAg only dropped below the level observed in control mice when the mice had received siPD-L1 during prime but not during boost vaccination (Figure 4B). This was accompanied by a significant reduction of pgRNA in the liver (Figure 4D), indicating an improved clearance of HBV by cytotoxic T cell activity. Taken together, these data indicate that checkpoint blockade during priming of T cell responses is essential to activate the full antiviral activity of virus-specific T cells.

## 4. Discussion

Our data suggest that the siRNA-mediated PD-L1 blockade can enhance the efficacy of therapeutic vaccination and may even be superior to the effects of an antibody against PD-L1 in terms of activating T cells locally in the liver. We demonstrate that therapeutic vaccination in combination with RNAi-mediated checkpoint inhibition improves the HBV-specific T cell response, is effective in reducing viral antigens in the blood, and is able to control HBV replication in and clearing HBV-positive hepatocytes from the liver. Here, checkpoint inhibition during priming of T cells proved more effective than during expansion by the MVA vector-boost vaccination. An siRNA-mediated knock-down of PD-L1 on its own did not show any beneficial effect despite still bearing a risk of liver toxicity. Because the combination of therapeutic vaccination with PD-L1 down-regulation by siRNA can enhance T cell cytotoxicity, potential liver toxicity needs to be carefully monitored. 

Despite substantial efforts, it has rarely been achieved to restore HBV-specific immunity by therapeutic vaccination [11]. This may on the one hand be due to a suboptimal vaccine design. On the other hand, a number of parameters typically found in chronic hepatitis B are correlated with a poor outcome of immune therapies. These include long-term exposure to high antigen levels, overexpression of immune inhibitory molecules, and an unfavorable metabolic environment in the liver [22]. We have previously shown that reducing HBV antigen levels in mice improves *TherVacB* efficacy [8], and that the effector function of induced T cells strongly correlates with PD-1 expression on their surface. Based on these findings, the effect of PD-L1 suppression was investigated in this study. Although the idea of PD-1/PD-L1 pathway inhibition in the context of therapeutic vaccination is not new [5], it is usually achieved by using antibodies against PD-1. Here we demonstrate that using siRNA to knock-down critical immune checkpoint molecules and thus modify the microenvironment in the target organ, the liver, is a feasible way to achieve an organ-specific checkpoint inhibition. 

Antibody-mediated interference with immune checkpoints represents a systemic treatment that as such harbors the risk of general side effects frequently observed in cancer patients [16]. Targeting immune checkpoints in a distinct organ such as the liver is a promising means to reduce such systemic side effects. In our study, a transfection reagent was used that allows a hepatocyte-specific targeting of siRNA, which proved sufficient to limit the immune suppression in the liver. With recent advances in siRNA and other non-coding RNA therapies and the development of site-specific transfection and delivery reagents, new opportunities and more sophisticated, targeted treatment approaches are expected to become available in the future. A prominent example which is already used in the clinics is the coupling of siRNAs to *N*-Acetylgalactosamine (GalNAc) that allows efficient siRNA delivery into hepatocytes following subcutaneous injection [18]. Further characterization of the distribution of PD-L1 and other co-inhibitory molecules in the liver, and reagents that allow targeting of distinct liver cell populations such as sinusoidal endothelial cells, stellate cells or liver macrophages, may lead to even more sophisticated inhibition of disease progression by only targeting those cells which are subject to the HBV-related immune suppression or, e.g., relevant to fibrosis development. 

Despite a slight ALT elevation in mice receiving siRNA against PD-L1, the siRNA treatment was well tolerated and when combined with *TherVacB* highly efficient in augmenting T cell functionality and antiviral efficacy, allowing to control HBV replication and even achieve clearance from the liver. ALT elevation can either be due to siPD-L1 toxicity or due to an activation of potentially auto-aggressive T cells in the liver, or to the increased cytotoxic activity of HBV-specific T cells after therapeutic vaccination. As we once observed a transient ALT elevation when applying siPDL-1 without concurrent therapeutic vaccination, we assume that siPDL-1 treatment per se may be able to cause side effects comparable to what is observed after antibody-mediated checkpoint inhibition. This may be further enhanced by HBV-specific T cell effector function after therapeutic vaccination. Further studies will be required to determine whether checkpoint inhibition by antibodies or by RNAi is better tolerated. Timepoint-dependent differences in the efficacy of checkpoint inhibition point at the need to further characterize the PD-1/PD-L1 pathway in vaccination-induced T cell responses. Earlier studies highlighted that timing and context may influence the outcome of the effector response when blocking immune checkpoints [23]. In contrast to their findings, we observed a lack of any effect when checkpoint inhibition was applied during boosting of T cell responses by MVA. Overall, however, this demonstrates the delicate balance of checkpoint inhibition between preventing autoimmunity and enhancing pathogen-specific immune responses. When translating these findings into the clinics, side effects and potential liver toxicity need to be closely monitored, and a careful approach must be taken to find an effective dose without disturbing the immune system’s homeostasis.

## Figures and Tables

**Figure 1 biomolecules-12-00470-f001:**
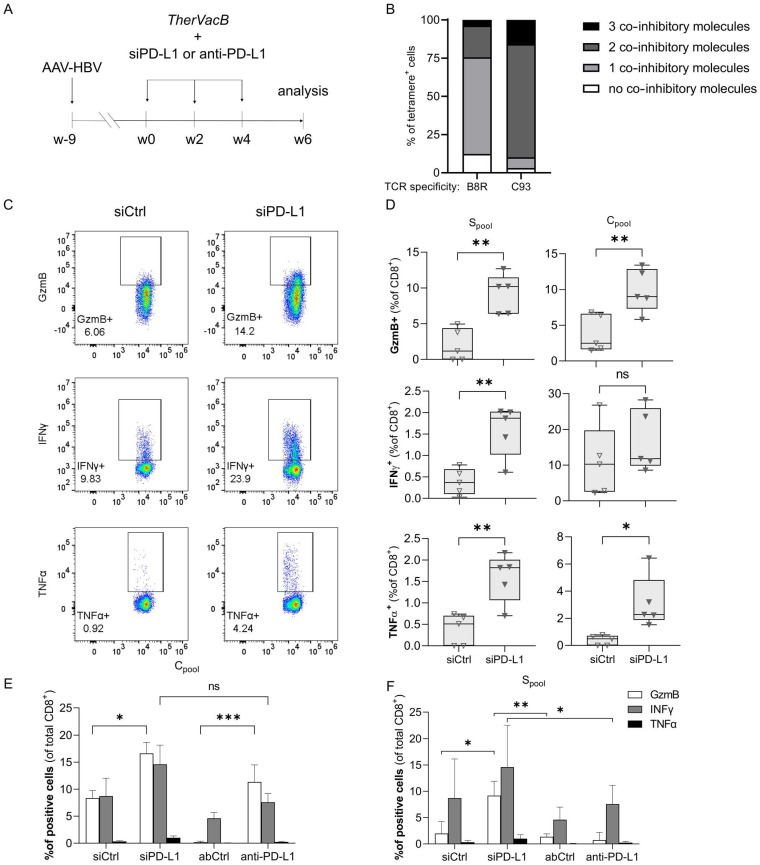
Effect of PD-L1 checkpoint blockade on functionality of HBV-specific T cells. (**A**) C57Bl/6J mice (*n* = 5 per group) were infected intravenously with AAV-HBV resulting in persistent HBV replication. After nine weeks, mice were vaccinated two times using a combination of HBc and HBsAg adjuvanted with c-di-AMP (week 0 and week 2), and one time using an MVA expressing HBV core and S (week 4). Either 1 mg/kg siRNA or 100 µg antibody was administered i.v. one day prior to each vaccination. Mice were sacrificed at week 6. (**B**) HBV-specific (C_93_) and MVA-specific (B8R) liver-associated lymphocytes (LAL) identified by multimer staining were co-stained for PD-1, TIM3 and LAG-3. The percentage of cells expressing one or more of these markers is indicated. (**C**–**F**) Liver-associated lymphocytes were isolated and restimulated for 16 h with peptide pools covering the HBV S- (S_pool_) or core protein (C_pool_) before intracellular cytokine staining of granzyme B (GzmB), IFNγ and TNFα was performed. (**C**) Representative flow cytometry stainings of LAL from single animals after stimulation with C_pool_ are shown. (**D**) Percentage of CD8^+^ LAL staining positive for the indicated cytokine per animal. Box plots show median, interquartile ratio (IQR) (box), and minimum to maximum (whiskers). (**E**,**F**) Five mice per group received either control or PD-L1-specific siRNA or antibody as indicated. LAL were isolated on day 14 after the last vaccination, stimulated with (**E**) S_pool_ or (**F**) C_pool_ and intracellular cytokine staining was performed. Mean ± SEM of the percentage of CD8^+^ T cells staining positive for the indicated cytokine are shown. Data from five animals per group are shown. Welch’s *t*-test or repeated measure two-way ANOVA with Tukey’s multiple comparison correction was performed; * = *p* < 0.05, ** = *p* < 0.01. *** = *p* < 0.001.

**Figure 2 biomolecules-12-00470-f002:**
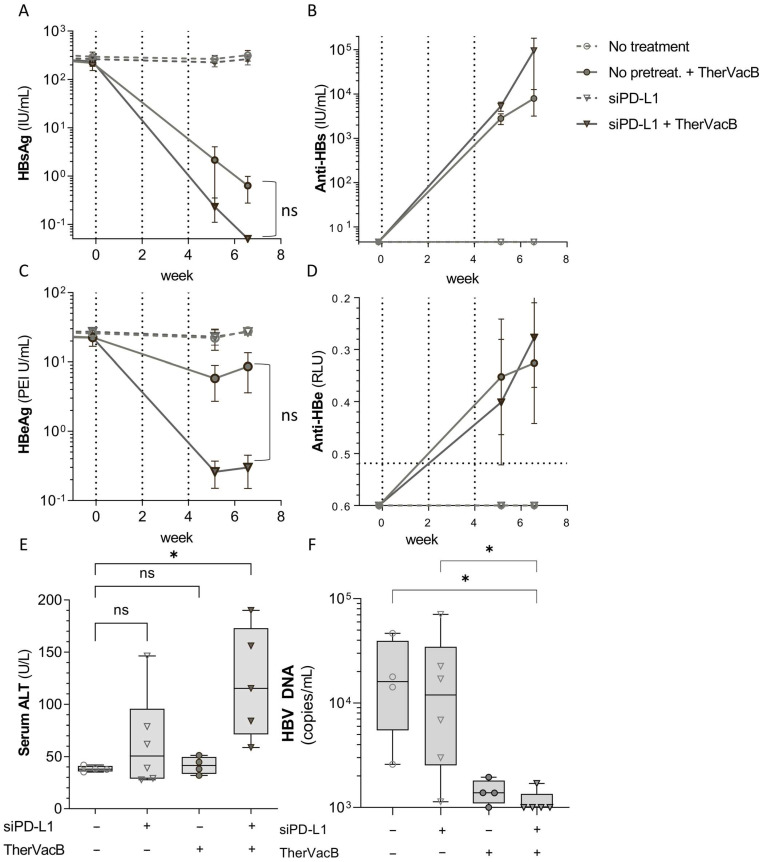
Antiviral efficacy of a combination of siPD-L1 and *TherVacB*. (**A**–**E**) Bl6 mice were infected intravenously with AAV-HBV resulting in persistent HBV replication. After nine weeks, mice were vaccinated two times using a combination of HBc and HBsAg adjuvanted with c-di-AMP (week 0 and week 2), and one time using an MVA expressing HBV core and S (week 4) (arrows in the graph indicate vaccinations); 1 mg/kg siRNA was administered i.v. one day prior to each vaccination. Mice were sacrificed at week 7 after start of vaccination. (**A**) HBsAg, (**B**), anti-HBs (**C**) HBeAg and (**D**) anti-HBe were measured in at indicated timepoints (arrows in the graph represent timepoints of vaccination). (**E**) Serum ALT activity was measured at the final timepoint. (**F**) Serum HBV DNA was measured using RT-qPCR at the final timepoint. (**A**–**D**) Graphs show mean and standard error of the mean (whiskers). Statistical analyses were performed using two-way ANOVA with Bonferroni correction for multiple comparison. (**E**) Box plots show median, interquartile ratio (IQR) (box), and minimum to maximum (whiskers). Data from six animals per group are shown. Repeated measure one-way ANOVA with Dunnett’s multiple comparison correction; * *p* < 0.05.

**Figure 3 biomolecules-12-00470-f003:**
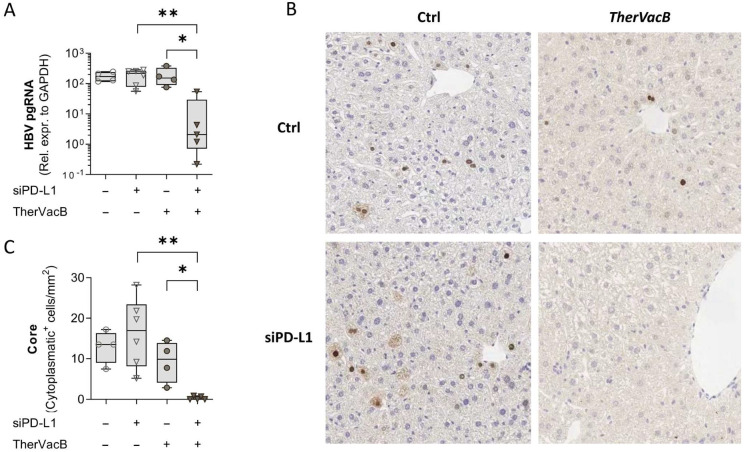
Effect of *TherVacB* and siPD-L1 on HBV persistence in the liver. (**A**) HBV pregenomic RNA (pgRNA) was quantified in liver lysates by RT-qPCR. (**B**,**C**) Quantification of hepatocytes that stained positive for HBc using immunohistochemistry. (**B**) One representative staining per group is shown. (**C**) Box plots show median, interquartile ratio (IQR) (box), and minimum to maximum (whiskers) numbers of cells with HBc^+^ cytoplasm per animal. (**A**,**C**) Data from four to six animals per group are shown. Kruskal–Wallis with Dunn’s multiple comparison correction; * *p* < 0.05, ** *p* < 0.01.

**Figure 4 biomolecules-12-00470-f004:**
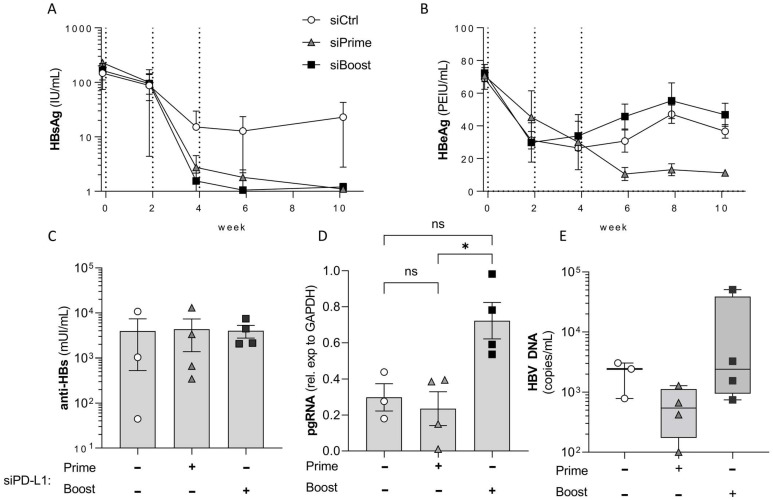
PD-L1 knock-down in the liver during prime, but not during boost vaccination enhances the effect of therapeutic vaccination. Bl6 mice (*n* = 3 to 4 per group) were infected intravenously with AAV-HBV to induce persistent HBV replication. After eight weeks, mice were vaccinated two times using a combination of HBc and HBsAg adjuvanted with c-di-AMP for prime vaccination (week 0 and week 2), and then boosted once using an MVA expressing HBV core and S (week 4); 1 mg/kg siRNA was administered i.v. one day prior to vaccination either twice before the prime vaccinations (siPrime) or once before the MVA boost (siBoost). Mice were sacrificed at week 10 after start of vaccination. Serum levels of HBsAg (**A**) and HBeAg (**B**) were measured at indicated timepoints. Dotted vertical lines indicate the three vaccinations. (**C**) Anti-HBs was quantified at the final timepoint. HBV pgRNA (**D**) in the liver and DNA (**E**) in the serum were quantified and measured by RT-qPCR. Repeated measure two-way ANOVA with Tukey´s multiple comparison correction; * *p* < 0.05.

**Table 1 biomolecules-12-00470-t001:** Primers to quantify HBV pgRNA.

Primer	Target	Sequence (5′ to 3′)
HBV3.5 kbRNA_fw ^a^	HBV 3.5 kb RNA	GAGTGTGGATTCGCACTCC
HBV3.5 kbRNA_rev ^a^ mGAPDH_fw ^b^ mGAPDH_rev ^b^	HBV 3.5 kb RNA Murine *gapdh* gene Murine *gapdh* gene	GAGGCGAGGGAGTTCTTCT ACCAACTGCTTAGCCC CCACGACGGACACATT

^a^ Cycling conditions: 5 min 95 °C followed by 40× (15 s 95 °C, 30 s 60 °C). ^b^ Cycling conditions: 5 min 95 °C followed by 40× (15 s 95 °C, 10 s 60 °C, 25 s 72 °C).

## Data Availability

The data presented in this study are available on appropriate request from the corresponding author. The data are not publicly available to ensure IP protection.

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
