# Peer review of "PD-L1 Silencing in Liver Using siRNAs Enhances Efficacy of Therapeutic Vaccination for Chronic Hepatitis B"

_biomolecules, 2022, doi:10.3390/biom12030470_

Round 1

Reviewer 1 Report

In this paper, Bunse et al. analyzed the effects of siRNA against PD-L1 on the efficacy of therapeutic vaccination using a mice model with AAV encoding HBV genome. Silencing of PD-L1 enhanced the efficacy of vaccination leading to a reduction of viral antigens. The results are promising for the establishment of novel antiviral strategy, but there are some issues to be addressed as below.

  1. Fig. 1A seems to indicate that AAV-HBV was injected 4 weeks before the vaccination as “W-4”, but it is described on line 146 that nine weeks after transduction, mice were immunized. Please show the exact schedule including timings of administration of vaccination, siRNA and antibody against PD-L1 in Fig. 1A.
  2. Fig. 2A-D – please show what dots and error bars indicate in the figure legend. Isn’t there really a significant difference of HBeAg between “no pretreat + TherVacB” and “siPD-L1 + TherVacB” ?
  3. Fig. 2 – in addition to viral antigens, HBV DNA levels in the serum samples should be analyzed. Similarly, HBV DNA levels should be shown in the analysis of Fig. 4.
  4. Fig. 4 – a group with siBoost shows higher levels of HBeAg and pgRNA than other groups. Please discuss the mechanisms of this result.

Author Response

  • Fig. 1A seems to indicate that AAV-HBV was injected 4 weeks before the vaccination as “W-4”, but it is described on line 146 that nine weeks after transduction, mice were immunized. Please show the exact schedule including timings of administration of vaccination, siRNA and antibody against PD-L1 in Fig. 1A.

The original vaccination scheme was used in the previous Fig, we corrected the timepoints according to our actual experiment records

  • Fig. 2A-D – please show what dots and error bars indicate in the figure legend. Isn’t there really a significant difference of HBeAg between “no pretreat + TherVacB” and “siPD-L1 + TherVacB” ? 

We added the missing Information regarding the dots and error bars and the statistical analysis used to determine statistical significance. Since only 3 out 5 mice achieved viral clearance in the “no pretreatment  + TherVacB” Group, this group has a high standard variation in both viral antigens measured in the serum and therefore the difference in the means compared to “siPD-L1 + TherVacB” is not significant.

  • Fig. 2 – in addition to viral antigens, HBV DNA levels in the serum samples should be analysed. Similarly, HBV DNA levels should be shown in the analysis of Fig. 4.

We analysed HBV DNA levels in the serum and added the results according to the reviewers suggestion to Fig.2 and Fig.4.

  • Fig. 4 – a group with siBoost shows higher levels of HBeAg and pgRNA than other groups. Please discuss the mechanisms of this result.

We are not sure why HBV-RNA levels increased in this group and can only speculate that HBV transcriptional activity may have increased due to the treatment or its effects. However, HBV-DNA levels are comparable to those without siRNA treatmentalthough HBV RNA levels increased after siBoost.

Reviewer 2 Report

The paper is interesting and well written. I draw the authors' attention to one small error, and I have a question for them:

  1. Lines 223-224: In the description of Figure 2, the information is mistakenly exchanged: „anti-HBe (C) and anti-HBs (D)“. The labels (C) and (D) should be reversed.
  2. Lines 259-263: Mice that received si-PD-L1 during booster vaccination (siBoost) cleared HBsAg but not HBeAg. How do you explain the continued production of HBeAg in this group of mice, i.e. in the group of HBsAg negative mice where we assume the absence of HBV replication?

Author Response

  • Lines 223-224: In the description of Figure 2, the information is mistakenly exchanged: „anti-HBe (C) and anti-HBs (D)“. The labels (C) and (D) should be reversed.

Fig. legend was corrected accordingly.

  • Lines 259-263: Mice that received si-PD-L1 during booster vaccination (siBoost) cleared HBsAg but not HBeAg. How do you explain the continued production of HBeAg in this group of mice, i.e. in the group of HBsAg negative mice where we assume the absence of HBV replication?

This is a phenomenon which we and other typically observe (e.g. Michler, Kosinska et al. Gastroenterology 2020). Prime immunization with protein induces anti-HBs antibodies which complex HBsAg so that it cannot be detected anymore. Clearence of HBeAg, in contrast, requires cytolytic T cell function and elimination of hepatocytes. This is only achieved when T cells become activated by MVA immunization.

Reviewer 3 Report

Bunse et al. studied the effects of PD-L1 siRNA in combination with a heterologous prime-boost therapeutic vaccination scheme in the AAV-based mouse model of chronic HBV infection. They found that si-PD-L1 treatment with TherVacB resulted in a higher functionality of HBV-specific CD8+ T cells (with increased GzmB, IFNgamma, TNFa), as well as decreased HBsAg and HBeAg levels. Interestingly, PD-L1 knock-down elicits the effects during prime but not during boost vaccination. Overall, this strategy is promising to treat chronic hepatitis B and has lower systematic side effects. Several minor issues.

  1. In lines 30-31, NUC treatment somehow reduces the risk to develop cirrhosis and HCC. The author's statement was confusing, which sounds like the NUCs treatment causes the risk of cirrhosis and HCC. It's better to make it clear.
  2. line 201, a typo "weather" should be "whether".

Author Response

  • In lines 30-31, NUC treatment somehow reduces the risk to develop cirrhosis and HCC. The author's statement was confusing, which sounds like the NUCs treatment causes the risk of cirrhosis and HCC. It's better to make it clear.

We corrected this statement.
 Original:” Current standard of care using nucleoside analogues suppresses HBV replication, but seldomly eradicates the virus, leaving patients at risk to develop liver cirrhosis and hepatocellular carcinoma [2].

New: „Current standard of care using nucleoside analogues suppresses HBV replication, but seldomly eradicates the virus, and although the patients at risk to develop liver cirrhosis and hepatocellular carcinoma is reduced it remains high compared to healthy controls  [2]. “

  • line 201, a typo "weather" should be "whether"

Thank you, the typo was corrected

Round 2

Reviewer 1 Report

The authors modified the manuscript and it has been improved appropriately. Please correct a typo on line 344 "PL-L1".

Author Response

Thank you for your comments. We have revised this typo.